# Virulence Factors of Enteric Pathogenic *Escherichia coli*: A Review

**DOI:** 10.3390/ijms22189922

**Published:** 2021-09-14

**Authors:** Babak Pakbin, Wolfram M. Brück, John W. A. Rossen

**Affiliations:** 1Institute for Life Technologies, University of Applied Sciences Western Switzerland Valais-Wallis, 1950 Sion 2, Switzerland; b.pakbin@ut.ac.ir; 2Department of Medical Microbiology and Infection Prevention, University Medical Center Groningen, University of Groningen, 9713 GZ Groningen, The Netherlands; john.rossen@gmail.com; 3Medical Microbiology Research Center, Qazvin University of Medical Sciences, Qazvin 15315-3419, Iran

**Keywords:** enteric pathogenic *Escherichia coli*, *E. coli* pathotypes, virulence factor genes

## Abstract

*Escherichia coli* are remarkably versatile microorganisms and important members of the normal intestinal microbiota of humans and animals. This harmless commensal organism can acquire a mixture of comprehensive mobile genetic elements that contain genes encoding virulence factors, becoming an emerging human pathogen capable of causing a broad spectrum of intestinal and extraintestinal diseases. Nine definite enteric *E. coli* pathotypes have been well characterized, causing diseases ranging from various gastrointestinal disorders to urinary tract infections. These pathotypes employ many virulence factors and effectors subverting the functions of host cells to mediate their virulence and pathogenesis. This review summarizes new developments in our understanding of diverse virulence factors associated with encoding genes used by different pathotypes of enteric pathogenic *E. coli* to cause intestinal and extraintestinal diseases in humans.

## 1. Introduction

*Escherichia coli*, rod-shaped and gram-negative bacteria belonging to the Enterobacteriaceae family, were first isolated from infant stool and characterized by Theodor Escherich in 1885 [1]. A few hours after birth, *E. coli* colonize and inhabit the gastrointestinal tract of infants. Regarding several mutual benefits, humans and commensal *E. coli* strains coexist without any adverse effects. However, commensal *E. coli* may cause disease in patients with breached gastrointestinal barriers or immunocompromised hosts [2]. Notably, certain *E. coli* mediate various diseases, including intestinal and extraintestinal disorders in humans and animals worldwide. Nine pathovars have been described for *E. coli* strains isolated from humans causing diarrheagenic and extraintestinal diseases [3]. Of these, seven pathotypes have been described as enteric pathogenic *E. coli,* including Enteropathogenic *E. coli* (EPEC), Enterohaemorrhagic *E. coli* (EHEC), Enterotoxigenic *E. coli* (ETEC), Enteroinvasive *E. coli* (EIEC), Enteroaggregative *E. coli* (EAEC), Diffusely adherent *E. coli* (DAEC), and, a new pathotype, Adherent-Invasive *E. coli* (AIEC), causing mostly diarrhea and intestinal disorders. However, the EHEC pathotype has been implicated in extraintestinal diseases such as Hemolytic Uremic Syndrome (HUS) [4]. Many of these pathotypes constitute public health concerns as foodborne pathogens and caused several fatal outbreaks in developing and developed countries [5,6]. Enteric *E. coli* pathotypes are implicated in many diseases through distinctly different pathogenesis. Pathogenesis is the process by which pathogens cause disease or disorder, often by expressing virulence-factor-encoding genes [7,8].

Virulence factors are specific molecules, primarily proteins produced and released by bacteria, fungi, protozoa, and viruses. These factors are encoded by specific genes located on the chromosome or mobile genetic elements (e.g., plasmids or transposons) in bacterial pathogens [9,10]. Each *E. coli* pathotype has its characteristic pathogenicity mechanisms and a specific profile of virulence factors encoded by specific gene clusters. Genes associated with pathogenicity may encode activities such as adhesion, invasion, attachment, iron acquisition, motility, and toxin activity, among others. We can distinguish four main virulence classes of *E. coli* pathotypes: colonization, fitness, toxins, and effectors; each consists of several specific virulence factors with a definite function and activity (Table 1). It is noteworthy that different enteric and extraintestinal pathotypes of *E. coli* isolates share the same virulence factors and strategies [11]. Many virulence factors associated with enteric *E. coli* pathotypes implicated in intestinal and extraintestinal disorders have been identified in the last years [12]. The investigation of these virulence factors and the associated encoding genes can provide explicit knowledge about the interaction between these factors in enteric *E. coli* pathotypes and host proteins at the molecular level, indicating how they lead to diseases and can implement preventive strategies against them [13]. This review summarizes recent advances in our understanding of virulence factors and the associated genes in enteric pathogenic *E. coli*.

## 2. Enteropathogenic *Escherichia coli* (EPEC)

EPEC is the principal cause of diarrheal diseases and outbreaks in infants characterized for the first time in the UK in 1945, with a high morbidity and mortality rate in children under six months of age. The EPEC rate in developed countries has decreased in recent decades. However, it is one of the major public health concerns for infants and adults in developing countries [14]. Some pathogenesis mechanisms of EPEC, such as the attaching and effacing (A/E) virulence factors, are shared with pathogens such as rabbit-EPEC, *Citrobacter rodentium* in mice, EHEC, and *Escherichia albertii* [15]. Foodstuffs, particularly milk and ground beef, are a vehicle of EPEC transmission to humans and animals and lead to intestinal infections [16,17,18]. 

Histopathologically, EPEC belong to the A/E pathogens. The main virulence-factor-encoded genes of EPEC are located on a 35.62 kb chromosomal pathogenicity island (PAI) within 41 open reading frames (ORFs) and a pathogenic plasmid-termed locus of enterocyte effacement (LEE) and EPEC adherence factor (EAF), respectively [19]. LEE gene expression is modulated by the LEE-encoded regulator gene, located on the first part of the LEE. Another PAI of EPEC is termed the *EspC* island, encoding a serine protease autotransporter enterotoxin gene with the same name, and this is the only toxin released by EPEC, which can lead to epithelial cell necrosis [14]. Regarding the EAF plasmid presence/absence, EPEC is classified into typical (t-EPEC) with complete virulence genes and atypical EPEC without EAF-harboring operon for bundle type IV pili (*bfp*) [20]. The most prominent virulence factors of EPEC in both typical and atypical pathogens are located on the LEE. This PAI is a type-three secretion system (T3SS) comprising the *EspA*, *EspB*, and *EspD* (EPEC secreted protein) translocator proteins, effector proteins (*Map*, *EspF*, *EspG*, *EspH*, *EspZ*), the proteins involved in intimate adherence, and regulatory elements [21]. Gut microbiome surface molecules, oxygen levels, and hormones in the environment mediate the signaling of LEE-activating transcription. A/E is initiated with the reaction between the outer membrane intimin and the translocated intimin receptor encoded by *eae* and *tir* genes, respectively [22]. Before T3SS expression following a stable attachment by intimin-Tir interaction and effectors injection, EPEC sticks to the surface of the host cells (initial attachment) using pili-like structures [23]. Type-IV pili (*bfp*) in a-EPEC and the lymphocyte inhibitory factor (lifA/efa1) in t-EPEC mediate the initial attachment to the enterocytes located in the small bowel considering the first step of A/E following the formation of localized microcolonies. T3SS is a complex injectisome machine in molecular nano-scale including more than 20 main proteins and consisting of three principal extracellular appendages: the needle, filament, and translocation pore; structurally, it is composed of a syringe with a central channel embedded into the bacterial membrane by ring structures and a supramolecular structure on the top of the needle connected to the host cell [24]. *EspA*, *espB*, and *espD* proteins construct different translocator structures of T3SS, and these proteins are cleaved by *espP* and *espC* [25]. After the translocation of Tir, a strong attachment is formed while the translocated protein interacts with intimin. The first result of the Tir–intimin interaction is the induction of a cytoskeletal rearrangement and the pedestal formation underneath the pathogen attachment side in the host cell [26]. The translocated Tir protein is phosphorylated by tyrosine kinase in the host cell. The phosphorylation of Tir recruits the NcK protein leading to the activation of the neural Wiskott–Aldrich syndrome protein (N-WASP), the stimulation of the actin-related protein (ARP) 2/3, and the mediation of actin polymerization and rearrangement, which results in pedestal formation. This cytoskeletal rearrangement contributes to immune modulation and diarrhea. A thick biofilm formation also has a crucial role in EPEC pathogenesis [27]. The activation of biofilm formation is mediated by Quorum-sensing encoded by the suppressor of the division inhibitor (*SdiA*) gene. However, for biofilm formation, bacteria demand structures to support these characteristic curli fimbriae, type-I fimbriae, and the cellulose encoded in EPEC by csgA, fimA, and bcsA genes, respectively [28].

The next step in EPEC pathogenesis is the translocation of LEE and Nle effectors and proteins into the host cell through the T3SS. Some of these effectors efface the host cell with multiple functionalities. In addition to attachment and actin pedestal formation, Tir inhibits the NF-kB signaling as the immune evasion mechanism in EPEC pathogenesis [29]. The mitochondrial-associated protein (MAP) shares the Trp-xxx-Glu motif conferring GTPase GEF activity. GEF induces the Cdc42 factor and contributes to filopodia formation at the bacterial attachment site. MAP also disrupts the mitochondrial membrane functionality, contributing to the host cell’s death. Another prominent effector is NleA (espI), having many roles, as proceeding with the NLRP3 inflammasome activation, tight junction disruption, and inhibition of the secretion of host cell cytokines [30]. EspF is a multi-function effector inducing mitochondrial death, disrupting the tight junction integrity, and implicated in phagocytosis inhibition, contributing to the immune evasion of EPEC during the pathogenesis. *EspB*, *H*, and *J* genes also expressed proteins to inhibit phagocytosis [31]. Another multi-functional non-LEE effector is *nleF*, which activates the inflammasome by binding to caspase-4, dampening the cellular immune response. *EspF*, *espT*, *MAP*, *nleF*, and the cycle-inhibiting factor (Cif) induce host cell death by blocking the progression of the cell cycle. Despite the identification of the functions of these effectors, multi and single effects of many *Nle* and *esp* proteins secreted by EPEC and other A/E pathogens have not been characterized yet (Table 1) [32].

## 3. Enterohaemorrhagic *Escherichia coli* (EHEC)

EHEC cause diarrhea, hemorrhagic colitis (HC) with bloody diarrhea, and hemolytic uremic syndrome (HUS) in humans and are implicated in several foodborne outbreaks in developed countries. The main reservoir of this pathogen is the intestinal tract of cattle. EHEC was first isolated from a patient with bloody diarrhea and gastrointestinal disorder in 1982 and led to a worldwide pandemic [33]. EHEC as foodborne A/E pathogens are mainly transmitted to humans through contaminated food and water [34]. Outbreaks of EHEC serotypes usually have occurred via the fecal-oral route by person-to-person transmission, animal contact, and consumption of under-thermally processed foods as well as undercooked meat products, unpasteurized apple juice, raw milk, or cross-contaminated raw vegetables such as lettuce and bean sprouts. Raw flour has also been recently identified as a source of EHEC outbreaks [8,35,36,37].

The most important serotype playing a role in EHEC outbreaks is O157: H7, which is still considered a serious health concern in Japan, Europe, and North America. EHEC serotype O104: H4 was first isolated from a gastrointestinal infection and HUS outbreak by the consumption of sprouts in Germany in 2011. From there, it was spread out around the world. However, the genome sequencing of this serotype revealed that it belongs to both EHEC and enteroaggregative *E. coli* (EAEC) pathotypes, introducing a new emerged pathotype of *E*. *coli* termed enteroaggregative hemorrhagic *E. coli* (EAHEC) [34,38,39].

The shiga-like toxin (SLT), also termed verotoxin and encoded by *stx* genes, is the main virulence factor in EHEC serotypes which belongs to the shiga-toxin producing *E. coli* group and is responsible for pathological manifestations leading to specific disease symptoms caused during EHEC infections, such as HUS and renal failure [8,40]. SLT includes two subgroups, *stx1* and *stx2*, and different subtypes. *Stx2a*, *stx2c*, and *stx2d* positive EHEC isolates are strongly associated with HC and HUS compared to other stx-subgroups and subtypes [41,42]. The shiga toxin is an AB_5_ toxin. Subunit A is cleaved into two fragments, A_1_ and A_2_, by reducing a disulfide bond, releasing the A_1_ fragment into the cytoplasm. It exerts the cytotoxic effect by enzymatically deadenylating position 4324 of 28s rRNA of the 60s ribosome, leading to the inhibition of protein synthesis and cell death [43]. In addition, to triggering ribotoxic stress, the A_1_ fragment of *stx* induces cytokine production and activates the apoptotic cell pathways. Protein translation inhibition and immune response modulation are mainly responsible for the pathogenesis of shiga toxin’s subunit A. The A subunit fragments need non-covalent binding with homopentameric B subunits to enter the target cell and induce the cytotoxic effect. B subunits of shiga toxins specifically bind to the carbohydrate moiety of glycosphingolipid globotriaosylceramide (Gb_3_ or CD77), a specific receptor molecule found on the surface of kidney epithelial and intestinal Paneth cells in humans. After receptor binding and Gb_3_-stx complex formation, the attached toxins form a cluster, membrane invagination, and endocytic pit formation at the plasma membrane of the cell [44]. These invaginations separate from the plasma membrane to form intracellular toxin carriers. Intracellular trafficking of the endocytic carriers is performed by moving from the early endosome to the Golgi and the endoplasmic reticulum before *stx* escapes from the Gb_3_-stx complex to exert the cytotoxic effect. *Stx* toxins are transported to the non-target (Gb_3_-negative) cells by neutrophil transmigration, micropinocytosis, and Gb_3_-independent transcytosis. Inside Gb_3_-negative cells, stx toxins only induce an inflammatory response and do not prevent protein synthesis [45]. There is a lack of a definite secretion system to release *stx* toxins by EHEC serotypes. Shiga toxins are released through phage-mediated lysis while specific triggers, including SOS-inducing agents such as antibiotic therapy or DNA damages, depress the transcription of lytic phase genes. This contributes to the lysis of the STEC bacterial cell and the release of shiga toxins into the extracellular milieu [46].

EHEC serotypes are a subgroup of STEC strains containing adhesin and attachment virulence factors encoded by the LEE pathogenicity island. Adhesin proteins are involved in the colonization and biofilm formation of EHEC on the abiotic and biological surfaces [47]. The first step of EHEC adherence to intestinal epithelial cells is initial attachment. Recent studies showed that the initial contact and attachment between the pathogen and the host cell are mediated through the interaction between the pathogen’s long polar fimbriae (*lfp* gene) and extracellular membrane proteins in the host cell, including fibronectin, collagen IV, and laminin [48]. They closely intimate the attachment, and the A/E effect occurs via the interactions of intimin (encoded by eae gene) and the receptor proteins in the host cell membrane, consisting of Tir (injected through T3SS), nuclein, and β1-integrins. The A/E pathogenic mechanism in EHEC is different from that in EPEC. Pedestal formation and actin rearrangement are mediated through an Nck-independent mechanism. Cell actin cytoskeleton rearrangement is induced while the Tir–intimin complex links to the *EspF* protein by a homologue protein of insulin receptor substrate [49]. N-WASP and ARP2/3 are activated for actin assembly via interaction with the induced *EspF*. Curli (encoded by *csg* genes) are recognized as adherence and colonization factors. Colonization, microcolonies, and biofilm formation are mediated by *E. coli* common pilus (ECP) and Hemorrhagic *E. coli* common pilus (HCP) via interaction with the surface of the host cell [49]. Some proteins, such as type 1, F9, and *E. coli* laminin-binding fimbriae are involved in the adhesion to the host cell. Autotransporters including *Eha*, *Saa*, and Sab proteins released by EHEC contribute to biofilm formation and adhesion [50]. Moreover, a plasmidal gene termed *ToxB*, located on the pO157 plasmid, may be involved in the adhesion in serotype O157: H7 and other EHEC strains. It is worthwhile to note that specific environmental conditions such as pH, temperature, and nutrient limitations induce the transcription and expression of adhesin genes through some signaling systems, such as the secretion of quorum-sensing molecules [51]. EHEC serotypes secrete twice as many effectors into the host cell via T3SS than EPEC. Host inflammatory responses induced by A/E, colonization, and biofilm formation also contribute to the symptoms of GI disorders. *NleF*, as a non-LEE effector, is considered a prominent virulence factor because of the counteraction to the host inflammatory response and inhibition of cell death. Several non-LEE-effectors such as *NleB*, *NleE*, *BleG*, *NleH*, and *EspJ* have been recently shown to mediate EHEC survival and biofilm formation (Table 1) [52,53,54].

## 4. Enterotoxigenic *Escherichia coli* (ETEC)

The *E. coli* pathotype ETEC releasing enterotoxins in the human small intestine is the principal cause of travelers’ and children diarrhea in developing countries. The highest infection and mortality rates of ETEC are found in children under the age of two. According to the World Health Organization’s reports, ETEC annually causes more than 157,000 human diarrheal cases that lead to death [55]. The main symptoms caused by ETEC infection are mild to severe diarrhea and abdominal pain. Other symptoms, such as vomiting, nausea, fever, and headache have rarely been observed [56]. ETEC are also animal pathogens causing acute diarrheal diseases in cattle, poultry, and piglets [57]. Generally, the two main virulence factors in ETEC that lead to diarrhea in humans are colonization factors and enterotoxins [58].

ETEC first adhere to the small intestine epithelial cells via surface structures and then secrete the enterotoxins. Colonization plays a fundamental role in the initial step of the pathogenesis mechanism of ETEC. ETEC engage with small bowel epithelial cells by various plasmid-encoded colonization factor (CFs) genes encoding fimbrial, non-fimbrial and fibrillar structures [59]. Among many CFs, CFA/I, CFA/II, and CFA/IV are the main factors to mediate the colonization of ETEC and have been highly detected in ETEC isolated from clinical samples. Other proteinaceous fimbrial and non-fimbrial structures include coli surface antigens (CS) such as CS1-CS6, type 1 fimbriae (*fimH*), *E. coli* common pili (*ecpA*) [60], autotransporter proteins such as the *Etp* group of proteins, and *EatA* (a member of SAPTE family). Flagella promote the initial attachment, adhesin, and colonization of ETEC to the host intestinal epithelial cells [61]. The *EtpA* secreted by ETEC is a glycoprotein containing N-acetylgalactosamine (NAG). NAG also presents as the sugar moiety on A blood group glycans. Therefore, ETEC contributes to a more severe diarrheal disease among humans with Type A blood [62].

The most prominent and effective virulence factor of ETEC is the secretion of enterotoxins. Two types of enterotoxins, heat-stable toxins (STs) and heat-labile toxins (LTs), are secreted by ETEC and activate cyclic nucleotide production, contributing to intestinal net water, salt, and fluid loss, causing secretory diarrhea in humans and animals. LTs, STs, or a combination of both toxins might be expressed by ETEC strains [63]. Genes encoding enterotoxins are located on both plasmids and the chromosome (prophages), which can be transferred among the different *E. coli* pathotypes. A 72 amino-acid (aa) protein (as the toxin precursor) is synthesized, of which a 19–18 aa region is released as the active form of the heat-stable toxin. These toxins are divided into STI and STII encoded by *estA* and *estB* genes, respectively located on plasmids. Considering the host species, STI is classified into STp (18 aa) and STh (19 aa) subtypes, mainly found in porcine and human ETEC strains, respectively. However, both subtypes have been detected in ETEC isolated from humans [64]. The structure of these toxins mimics the biologic function of mammalian peptides, including guanylin and uroguanylin, binds to the guanylate cyclase C receptor, and increases the intracellular cyclic guanosine monophosphate (cGMP) via the hydrolysis of guanosine triphosphate. The increase in cGMP concentration activates the protein kinase that phosphorylates the cystic fibrosis transmembrane regulatory (CFTR) channel. The activated CFTR channel inhibits ion exchange and sodium reabsorption, contributing to the release of water and salt into the intestinal lumen and thus net fluid loss, leading to watery and secretory diarrhea [65,66]. The putative export channel for the secretion of ST is an outer membrane protein encoded by the *tolC* gene. The *EtpA* adhesin factor gene, located on a plasmid, expresses a protein to construct a molecular bridge structure between bacteria and the host cell. Without the expression of *tolC* and *EtpA* genes, the delivery of ST to the target cells is not performed. These genes have also been considered the main virulence factors of ETEC [67]. Another mechanism of ST triggering diarrhea is the tight junction modulation of intestinal epithelial cells, leading to an increase in permeability by suppressing the expression of proteins and disturbing the integrity of the tight junction. In addition, ST toxins mediating the innate immune responses led to the secretion of chemokines and cytokines such as IL-6 and IL-8 [63].

Another toxin secreted by ETEC after colonization of the small intestine is a heat-labile toxin. Two types of LT have been identified, LT type I and II. LT (type I) is also categorized into LTh and LTp subtypes, detected in strains isolated from humans and pigs. However, LT II is mainly associated with strains isolated from animals and occasionally from humans [67]. LT is initially synthesized as a holotoxin in the periplasm and then released via two secretion systems: a type-II secretion system (T2SS) and outer membrane vesicles (OMV). ETEC mainly secretes LT through the OMV system. ST and LT toxins are composed of a single A subunit with enzymatic activity and a pentameric B subunit binding to the GM1 ganglioside receptor on the host cell surface [68]. The structure and pathogenesis mechanism of LT is identical to that (about 80%) of the cholera toxin secreted by *Vibrio* cells. After delivery and translocation into the host cell, the A subunit induces the ADP-ribosylation factor and inhibits GSα GTPase activity, leading to the activation of adenylate cyclase and the increase of intracellular cyclic adenosine monophosphate (cAMP) concentrations. Higher levels of cAMP contribute to the activation of the CFTR ion channel, followed by the secretion of water and electrolytes, leading to watery diarrhea [69]. By interacting with the carbohydrate-binding sites, LT also binds to blood groups A and B determinants. LT also facilitates the ETEC pathogenesis by improving the initial adherence, intestinal colonization, and increased virulence factors expression [64]. There are other virulence factors detected in ETEC strains isolated from humans with diarrhea. Secretion of an enteroaggregative heat-stable toxin (EAST1) by ETEC, encoded by the ast gene, with a similar function to the ST toxin, leads to increased intracellular levels of cGMP. The SPATE *EatA*, located on a large virulence plasmid, also increases the virulence of ETEC by facilitating the LT delivery (Table 1) [70].

## 5. Enteroinvasive *Escherichia coli* (EIEC)

Severe mucosal and bloody diarrhea with abdominal cramps and fever are typical clinical features of bacillary dysentery or shigellosis caused by different species of *Shigella* and EIEC as invasive foodborne pathogens [13,71]. Initially known as *Bacillus dysenteriae*, Shigella was previously characterized in 1897 by Kiyoshi Shiga in Japan after a severe epidemic causing more than 91,000 cases, with 20% mortality [72]. Fifty years later, EIEC was discovered and characterized with the same genetic, pathogenic, and biochemical properties as were observed for *Shigella.* Thus, they may be categorized in a single pathovar. However, a considerable distinction is still maintained regarding the clinical significance of *Shigella* species [1,73]. EIEC and *Shigella* are unable to ferment lactose, are non-motile, and lysine decarboxylase negative. These properties are considered to differentiate *Shigella* species and EIEC from other bacteria [74]. EIEC differs from other pathovars of *E. coli* because it is an obligate intracellular pathogen with neither adherence nor flagella factors [71]. *Shigella* comprise four species, including *S. sonnei*, *S. flexneri*, *S. boydii* and *S. dysenteriae*, which are responsible for varying degrees of diarrhea and acute intestinal and extra-intestinal diseases in humans [73,75,76]. EIEC strains primarily elicit watery diarrhea and cause invasive inflammatory colitis [77].

EIEC/*Shigella* infection commences with the penetration of the pathogen into the epithelial cells in the colon, passing through the microfold cells and reaching the underlying submucosa by a transcytosis mechanism [78,79]. The disruption and damage of tight junctions caused by inflammation also give EIEC access to the underlying submucosa [80]. EIEC is taken up by macrophages, where it lyses the endocytic vacuole and escapes from the phagosome. Following the release from dead macrophages, EIEC invades the basolateral sides of the colonocytes, survives and multiplicates intracellularly, moves directionally through the cytoplasm, and extends into the adjacent epithelial cell [13]. The virulence of this pathogen is primarily due to 220-kb plasmid-encoding virulence factors including the T3SS complex, chaperones, transcriptional regulators, translocators, and more than 25 effector proteins [4]. A T3SS needle, strongly required for the invasion, apoptosis of macrophages, and cell survival, is encoded on the mxi-spa locus of the virulence plasmid [11]. Before the invasion, the adhesion of EIEC to the epithelial cells is mediated via the binding of proteins encoded by the *IpaBCD* complex and *IpaB* genes to the α_1_β_5_ integrin and hyaluronan CD44 receptors, respectively. Membrane ruffling and epithelial cell cytoskeleton reorganization are mediated by *IpaA*, *IpaC*, *IpgB1*, *IpgD*, and *virA* to provide pathogen uptake [81]. *IpaH7.8*, *IpaB*, *IpaD*, and *IpaC* have also been implicated in phagosome escape [82]. *VirG* nucleates actin by inducing the acquisition of N-WASP, contributing to ARP2/3 formation and mediated unipolar actin tail formation on the surface of the bacteria [83]. *IpaC* also activates SRC-family protein kinases to recruit the ARP2/3 complex at the site of the bacterial contact and induces actin polymerization to form a ruffle for bacterial entry. *IpgB1* as mimicry of RhoG also promotes ruffle formation by the activation of RAC1. Actin microfilaments and their polymerization provide the propulsive forces needed for the directed movement of the pathogen throughout the host cell cytoplasm [80]. EIEC provides host cell integrity and survival in the early stages of infection by inhibiting intestinal epithelial cell detachment and turnover utilizing protein effectors *OspE* and *IpaB*, respectively. Some effectors, including *VirA*, *IpaA*, and *IpgD*, destabilize microtubules and actin and mediate the blockage of death in host cells to promote colonization and maintain its replicative niche [4,82].

Once localized in the epithelial cell cytoplasm, EIEC suppresses the host immune response and counteracts the immune defense system by using protein effectors to persist and survive inside the colonocytes [84]. *OspF* and *OspG* inhibit the activation and gene transcription of NF-кB. *IpaH*_9.8_ and *OspB* also suppress the expression of inflammatory cytokine levels such as interleukin-8. Other effectors, including *OspZ*, *OspI*, and *IpaH*_4.5_, also help EIEC downregulate and dampen the inflammatory responses [82]. Moreover, effectors *IcsB* and *VirA* promote host-cell-autophagy-mediated degradation in EIEC by sequestering the VirG and destabilizing microtubules, respectively [84]. Common clinical symptoms of watery diarrhea in EIEC and Shigella infections have been attributed to the additional virulence factors, including Shigella enterotoxins 1 and 2 encoded by ShET1 and ShET2 genes, respectively. ShET1 is located on the chromosome and a pathogenicity island, whereas ShET2 has been found in a virulence plasmid. Both enterotoxins contribute to secretary intestinal activity. ShET2 induces inflammation in the host intestinal epithelial cells [85]. Moreover, other toxins encoded by virulence factor genes such as *Pic*, *SepA*, *SigA*, and Sat are expressed and secreted in EIEC and Shigella species. Pic, encoded by chromosome locus, regulates ShET1 expression [80]. *SepA* and *Pic* toxin genes belong to the serine protease autotransporters of the Enterobacteriaceae (SPATEs) family toxin. SPATEs are protease-based toxins released by autotransporter pathways and secreted usually by pathogenic *E. coli* and *Shigella* species [86]. The *SigA*-encoded cytotoxin contributes to intestinal fluid accumulation in the EIEC or *Shigella*-infected hosts [86]. Recently, other virulence factors have also been detected and identified prevalently in EIEC and *Shigella* isolates, including *iutA*, *iucB*, *EatA*, *VirF*, *VirB*, *IpaJ*, and *OspC3* implicated for aerobactin synthesis, complex siderophore iron receptor, SPATE toxin, the regulation of virulence factor gene expression, the control of virulence factor gene synthesis, the inhibition of the host cell trafficking membrane, and the inhibition of inflammasomes, respectively (Table 1) [72,79].

## 6. Enteroaggregative *Escherichia coli* (EAEC)

EAEC has been considered an emerging foodborne pathogen primarily associated with persistent and acute childhood diarrhea and growth retardation in developed and developing countries [13]. Moreover, EAEC is the second leading cause of acute and persistent travelers’ diarrhea after ETEC in developed and developing nations, and one of the major causes of enteric infections in patients with HIV/AIDS [4,52]. Other symptoms of EAEC infection include vomiting, nausea, borborygmi, anorexia, and tenesmus [87,88]. As mentioned before, a hybrid strain of EAEC/STEC (serotype O104: H4) caused a large outbreak in 2011 in Germany, resulting in more than 4300 diarrhea cases and 50 deaths [89]. Several studies indicated that EAEC has the potential for efficient gut inflammation and enteric colonization, which might intensify the effects of other pathogenic bacterial virulence strategies [7]. EAEC often causes watery diarrhea, occasionally accompanied by blood or mucus. EAEC colonizes the mucosa of small and large bowels, contributing to mild to severe inflammation in the colon [90]. Traditionally, EAEC has been characterized by a “stacked-brick” pattern adherence to Hep-2 cells. Due to the heterogenicity of EAEC, the pathogenic mechanisms of this pathogen are very complicated. EAEC pathogenesis involves three steps: adherence to the intestinal epithelium via adherence aggregative fimbriae, biofilm formation, and secretion of toxins, mucosal inflammation, and cytotoxic damages [32,88,91].

The virulence factors of EAEC are encoded on a family of virulence plasmids, the plasmid of aggregative adherence called pAA, and pathogenicity islands distributed throughout the chromosome [92]. AggR is the main virulence factor regulator of EAEC. It is a member of a bacterial transcriptional regulator family, AraC, and is located on the pAA plasmid to control the expression of plasmid-borne and chromosomal virulence factor-encoded genes [93]. Epidemiological studies showed a significant association between the presence of the AggR gene, diarrheal diseases, and the concentrations of fecal cytokines in patients with diarrhea infected with EAEC strains [94,95,96]. Therefore, EAEC strains are divided into typical, carrying AggR, and AggR-regulated virulence factors and atypical EAEC, lacking the AggR regulon. The first stage of EAEC pathogenesis is the attachment and adherence to the intestinal mucosal cells [97]. The aggregative adherence of EAEC is associated with the aggregative adhesion fimbria (AAF), which is encoded by AAF genes [97]. Moreover, the initial attachment of EAEC to the epithelial cells is facilitated by AAFs. Five variants of AAFs have been identified, including AAF/I–V encoded by *aggA*, *aafA*, *agg3A*, *agg4A*, and *agg5A*, respectively. All of them are located on pAA. AAF genes have exclusively been detected in EAEC pathotypes [90]. The adherence mediated by AAFs and flagellin also induces cytokine responses [98]. The isoelectric point of AAFs is relatively high (pI: 8.9–9.4) compared with other adhesins. Throughout the gut (pH: 6–7.4), the AAFs bear a high positive charge [97]. Regarding the negative charge of lipopolysaccharide on the surface of gram-negative bacteria, an extension of AAFs away from the cell surface is mediated with the help of the secretion of a protein called dispersin. Dispersin is encoded by the aap gene located on pAA and can mask the negative charge conferred by lipopolysaccharides [99]. This protein counteracts excessive aggregation and causes the dispersion of EAEC across the mucosal epithelial cells. Other, less critical factors associated with EAEC adherence are alternative fimbrial structures, including type-IV pili [100]. Biofilm formation is another pathogenic mechanism of EAEC, and it is entirely distinct from that mediated by non-pathogenic *E. coli*. A type-VI secretion system (T6SS) encoded by *aaiA-Y* genes, located on a pathogenicity island identified on the chromosome and activated by AggR, mediates biofilm formation in EAEC. However, *aaiA* and *aaiC* genes have more commonly been found in typical EAEC strains worldwide [90,101].

The final stage of EAEC pathogenesis is toxin secretion, causing several effects, including enlarged crypt opening, microvillus vesiculation, and epithelial cell extrusion [88]. These putative toxins include a plasmid-encoded toxin, a protein involved in intestinal colonization, a *Shigella* extracellular enterotoxin, a secreted autotransporter toxin, and an enteroaggregative heat-stable toxin, which are encoded by the *pet*, *pic*, *sigA*, *sepA*, *sat*, and *astA* genes, respectively [90]. The *sigA* and *pic* genes are located on pathogenicity islands, and the other toxin genes are localized on pAA [102]. A thick mucus layer surrounds the EAEC biofilms, and the bacteria penetrate it through the secretion of the SPATE pic toxin with mucolytic activity [90]. *Shigella* extracellular enterotoxins (ShET1) are cytotoxin, enterotoxin, and IgA protease like homologues inducing intestinal cyclic GMP (cGMP) and cAMP-mediated secretion. This toxin has also been found in EAEC/STEC hybrid strains [88,103]. Pet, a 108 kDa protein belonging to the SPATE family, mediates the actin-binding protein fodrin, induces exfoliation, disrupts the actin skeleton, is trafficked through the endoplasmic reticulum, and induces the entry into the host cell. Pet also functions as a cytotoxin and heat-labile enterotoxin [104]. *Sat*, a SPATE toxin with cytotoxic and enterotoxic activities, impairs the tight junction and mediates autophagy in the epithelial cells. The enteroaggregative heat-stable toxin has mechanistic and physical similarities to the enterotoxin STa secreted by ETEC (Table 1) [105].

## 7. Diffusely Adherent *Escherichia coli* (DAEC)

DAEC is identified as a heterogenous pathotype of *E. coli* characterized by a diffuse pattern on Hep-2 and HeLa cells [88]. DAEC is associated with diarrhea in children between the ages of 1.5 and 5 years, urinary tract infection (UTI) in adults, pregnancy complications, and part of the intestinal commensal microflora in adults and children [106]. Watery diarrhea caused by DAEC can become more persistent in younger children. Adults are asymptomatic carriers of DAEC strains contributing to some chronic inflammatory intestinal diseases, including Crohn’s, coeliac, and inflammatory bowel diseases [107]. It has been shown that DAEC strains belong to the phylogenetic group B2, and it is worth noting that this phylogenetic group is predominant among the human commensal *E. coli* strains [32,88,108]. Despite the isolation of this strain from stools and duodenal cultures, none of the adult patients developed diarrhea [88]. The pathogenesis of DAEC starts with the attachment of the pathogen to specific host cells. The attachment of DAEC strains onto the urinary tract and enteric epithelial cells allows them to resist clearance by micronutrition and peristalsis, respectively [109]. The attachment also allows DAEC to interact with the host cell, release and deliver toxins, and trigger signaling events in host cells. DAEC attaches and adhere to the host cells using non-classical patterns and is mediated by Afa/Dr adhesins [110].

An important step in DAEC infection is the specific colonization of the small bowel mediated by the Afa/Dr family of adhesins, including the two main classes of adhesins presented on the surface of the pathogen: fimbrial adhesins (including linear polymers) and afimbrial adhesins (homotrimers or single proteins) [111,112,113]. Afa/Dr adhesins include Dr, DrII, AfaE/I–III, AfaE-V, NFA-I, and F1845 adhesins. Afa, Dr, and F1845 adhesins are encoded by *afaA-E*, *DraA-E*, and *daaA-E* genes, respectively [107,114]. F1845 and Dr adhesins bind to the decay-accelerating factor (DAF), a molecule expressed and found on the apical surface of urinary and intestinal epithelial cells, inducing cytoskeleton rearrangement and destroying microvilli [11,32,88]. The increased levels of DAF expression caused by the binding of Afa/Dr adhesins in patients with Crohn’s disease (CD) may contribute to the inflammation [110]. Some Afa/Dr adhesins can also bind to the epithelial cell surface receptor human carcinoembryonic antigen-related cell adhesion molecule (CEACAM), resulting in CDC42 activation, CEACAM aggregation at the site of bacterial adhesion, brush border microvilli effacement, and the mediation of a process leading to the microfilament-independent, microtubule- and lipid-raft-dependent internalization of the bacterial pathogen. These processes are thought to play a major role in the infection of intestinal epithelial cells by DAEC strains [115]. Moreover, Dr adhesins bind to type-IV collagen, necessary for DAEC-mediated urinary tract infections [88,115]. The interaction between Afa/Dr adhesins, flagella, and DAF receptors mediates IL-8 secretion and promotes the transmigration of polymorphonuclear neutrophils (PMNs). This causes DAF upregulation on the epithelial cell surface, providing more adhesion receptors [107,112]. The interaction between PMNs and Afa/Dr adhesins accelerates PMN-associated apoptosis and decreases the rate of phagocytosis mediated by PMNs. Across the epithelial cells, DAEC strains also induce the expression of MICA, which is potentially responsible for mediating inflammatory bowel disease (IBD) [112,116].

Although the pathogenesis of DAEC strains is mainly mediated through Afa/Dr adhesins, some SPATE toxins are secreted by this pathotype [112]. Afa/Dr DAEC strains release two main classes of SPATE toxins: class I, with cytotoxic activity against epithelial cells, including a secreted autotransporter toxin, a plasmid-encoded toxin, an extracellular serine protease, and *sigA* toxins, encoded by *sat*, *pet*, *EspP*, and *sigA* genes, respectively; class II, which are not cytotoxic and consist of a protein involved in intestinal colonization encoded by the *pic* gene [12,88,115]. In addition, other virulence factors, including type-I pili, pilus adhesin and enteroaggregative heat-stable toxins encoded by *pap*, *fim*, and *astA* genes, respectively, have been recently detected in DAEC strains (Table 1) [117].

## 8. Adherent Invasive *Escherichia coli* (AIEC)

AIEC is one of the most important causative agents of idiopathic inflammatory disorders, including IBD and CD, and primarily affect the human small bowel [118]. However, this pathotype may also be present as part of the normal microbiota of the human gut in healthy individuals and does not cause any disease [119]. No single distinct agent is defined as the leading cause of idiopathic IBD [120,121]. In addition to AIEC, other enteric pathogens have been reported to be involved in IBD and CD, including *Campylobacter* species, *Mycobacterium paratuberculosis*, and Cytomegalovirus [122,123]. The AIEC pathotype is an enteric pathogen able to adhere to and invade the intestinal epithelial cell layer and multiply within macrophages and epithelial cells. No particular virulence factors found in other *E. coli* pathotypes have been determined in AIEC strains [124,125]. AIEC pathogenesis consists of three stages: adhesion, invasion, and multiplication within the epithelial host cells. Initially, AIEC requires the adhesion to the epithelial host cells in the ileum, using the CAECAM6 expressed on these cells. Notably, the expression levels of CAECAM6 in CD patients is increased due to the stimulation of TNF-α production, causing intestinal inflammation by the colonization of AIEC [126]. Several virulence factors, such as outer membrane vesicles, outer membrane proteins, and long polar fimbriae, are expressed and utilized by AIEC strain after the adhesion to the epithelial host cells to mediate invasion, infect, and replicate within the macrophages [124,127].

Several virulence factors and associated encoding genes have been identified to play a role in the pathogenic interaction with the intestinal mucosa. Higher levels of expression of genes encoding virulence factors involved in adhesion, invasion, and survival encoding were detected in AIEC isolated from patients with CD, IBD, and ulcerative colitis [128,129,130]. Virulence factors present in AIEC isolated from patients with intestinal disorders include the adhesive subunit of type-1 fimbriae, the polypeptide stress response protein, the invasion protein *ibeA*, the invasion plasmid antigen, and virulence factors involved in ferric yersiniabactin uptake and capsule synthesis encoded by *fimH*, *yjaA*, *ibeA*, *ipaH*, *fyuA*, and *kpsMT* II genes, respectively [114,131,132]. AIEC strains have been considered a new therapeutic target for treating patients with IBD and CD [123]. However, several aspects of its pathogenesis are unknown and more genomic investigations, especially to determine the virulence factor-encoding genes, are required (Table 1) [118,127,129].

## 9. Conclusions

Enteric *E. coli* pathotypes display a high diversity of pathogenicity mechanisms and virulence factors profiles. They remain a significant concern in public health and food safety. The evolution of enteric *E. coli* pathotypes has contributed to the emergence of distinct *E. coli* pathotypes capable of toxin secretion, aggregative colonization, multiplying in the gastrointestinal tract, and damaging different environments through the adaptation of key genetic elements, resulting in the formation of new pathotypes [133]. The emergence of new *E. coli* pathotypes such as the hybrid EAEC/STEC (*E. coli* serotype O104: H4) strain that caused fatal outbreaks throughout Europe shows the importance of having a surveillance system in place. As food, water, animals, and people are potential vectors of enteric pathogenic *E. coli* transmission and contamination [5], such surveillance systems should be embedded in one-health approaches/networks. A wide range of functions including protein synthesis, specific gene transcription, the secretion of diverse micro/macro-molecules and ions, cytoskeleton rearrangement, apoptosis, autophagy, mitochondrial activities, cell division, and signal transduction in epithelial intestinal and extraintestinal host cells are affected by enteric *E. coli* pathotypes. This involves a broad spectrum of virulence factors encoded by specific gene clusters on the chromosome or mobile genetics elements. Therefore, for an optimal surveillance, new technologies such as high-throughput sequencing are required to gather precise information about the virulence profiles of *E. coli* pathotypes [134,135]. In addition, a comprehensive investigation of the role of *E. coli* pathotype virulence factors in pathogenesis will facilitate the development of novel therapeutics, the improvement of the design of effective vaccines, and the prevention of a further spreading of these diverse and widespread pathogens.

## Figures and Tables

**Table 1 ijms-22-09922-t001:** Virulence factor genes of enteric *E. coli* pathotypes: Colonization, fitness, toxins, and effectors.

Class	Virulence Factor	Activity/Function	Pathotype
**Colonization**	bfp	adherence,	EPEC
	eae	attaching and effacing of enterocyte,	EPEC, EHEC
	tir	translocated intimin receptor,	EPEC, EHEC
	lifA	initial attachment to the enterocytes,	EPEC
	csgA	curli fimbriae,	EPEC, EHEC
	fimA	type I fimbriae,	EPEC, EHEC, DAEC
	fimH	type I fimbriae,	ETEC, AIEC
	bcsA	cellulose structure,	EPEC
	eha	biofilm formation,	EHEC
	saa	biofilm formation,	EHEC
	sab	biofilm formation,	EHEC
	toxB	biofilm and adhesion establishment,	EHEC
	nleB	biofilm formation,	EHEC
	nleE	biofilm formation,	EHEC
	nleH	biofilm formation,	EHEC
	bleG	biofilm formation,	EHEC
	lfp	long polar fimbriae, initial attachment,	EHEC
	CFA/I	colonization factor,	ETEC
	CFA/II	colonization factor,	ETEC
	CFA/IV	colonization factor,	ETEC
	CS1-6	colonization (coli surface antigen),	ETEC
	etpA	initial attachment,	ETEC
	aggR	attachment and adherence,	EAEC
	aggA	aggregative adhesion fimbria,	EAEC
	aafA	aggregative adhesion fimbria,	EAEC
	agg3A	aggregative adhesion fimbria,	EAEC
	agg4A	aggregative adhesion fimbria,	EAEC
	agg5A	aggregative adhesion fimbria,	EAEC
	aap	dispersin, dispersion of EAEC,	EAEC
	afaA-E	Afa/Dr adhesins, IL-8 secretion,	DAEC
	DraA-E	cytoskeleton rearrangement, destroying microvilli, Afa/Dr adhesins, IL-8 secretion, expression of MICA	DAEC
	daaA-E	cytoskeleton rearrangement, destroying microvilli, Afa/Dr adhesins, IL-8 secretion, expression of MICA	DAEC
	pop	type I pili, adhesin,	DAEC
**Fitness**	sdiA	quorum sensing signaling,	EPEC
	iutA	aerobactin synthesis,	EIEC
	iucB	complex siderophore iron receptor,	EIEC
	yjaA	polypeptide stress response protein,	AIEC
	fyuA	ferric yersiniabactin uptake,	AIEC
	kpsMT II	capsule synthesis,	AIEC
**Toxins**	stx1	shiga toxin, surface localization of nucleolin and cytotoxic effect,	EHEC
	stx2	shiga toxin, surface localization of nucleolin and cytotoxic effect,	EHEC
	estA	ST I toxin, watery and secretory diarrhea, secretion of chemokines and cytokines,	ETEC
	estB	ST II toxin, watery and secretory diarrhea, secretion of chemokines and cytokines,	ETEC
	LT I	watery diarrhea,	ETEC
	LT II	watery diarrhea,	ETEC
	eatA	SPATE,	ETEC, EIEC
	astA	enteroaggregative heat-stable toxin, secretory diarrhea,	ETEC, EAEC, DAEC
	ShET1	shigella enterotoxin 1, secretary intestinal activity,	ETEC
	ShET2	shigella enterotoxin 2, secretary intestinal activity,	ETEC
	pet	SPATE, plasmid encoded toxin, inducing epithelial cell extrusion, host cell entering,	EAEC, DAEC
	pic	SPATE, ShET1 expression, inducing epithelial cell extrusion, mucolytic activity,	EIEC, EAEC
	sigA	SPATE, cytotoxin, accumulation of intestinal fluid,	EIEC, EAEC, DAEC
	sat	SPATE, secreted autotransporter toxin, impairing tight junction, mediating autophagy,	EAEC, DAEC
	sepA	shigella extracellular enterotoxin, cytotoxin, IgA protease like homologue,	EAEC
	hlyE	alpha hemolysin toxin,	EAEC
**Effectors**	espA	translocator structures of T3SS, E. coli common pili,	EPEC, ETEC
	espB	translocator structures of T3SS, phagocytosis inhibition,	EPEC
	espC	cleavage of T3SS translocator structures,	EPEC
	espD	translocator structures of T3SS,	EPEC
	espF	mitochondrial death, tight junction disruption, immune evasion, host cell death,	EPEC, EHEC
	espH	phagocytosis inhibition,	EPEC
	espJ	phagocytosis inhibition, biofilm formation,	EPEC, EHEC
	espP	cleavage of T3SS translocator structures,	EPEC, DAEC
	espT	host cell death,	EHEC
	MAP	disrupt mitochondrial membrane functionality, host cell death,	EPEC
	nleA	inflammasome activation, tight junction disruption, cytokines secretion inhibition,	EPEC
	etp	Autotransporter protein,	ETEC
	tolC	secretion of ST toxins,	ETEC
	nleF	host cell death, inflammasome activation,	EPEC, EHEC
	cif	cell cycle disruption, delays apoptosis,	EPEC
	ipaA	Type III effector, cytoskeleton reorganization, cell death blockage,	EIEC
	ipaB	Type III effector, adhesion, phagosome escape, cell turnover,	EIEC
	ipaC	Type III effector, adhesion, actin polymerization, phagosome escape,	EIEC
	ipaD	Type III effector, adhesion, phagosome escape, cell death blockage,	EIEC
	ipaH	dampen the inflammatory responses, invasion,	EIEC, AIEC
	ipaJ	inhibition of host cell trafficking membrane, inhibition of inflammasomes,	EIEC
	ipgB1	cytoskeleton reorganization, ruffle formation,	EIEC
	ipgD	cytoskeleton reorganization,	EIEC
	virA	cytoskeleton reorganization, cell death blockage, autophagy inhibition,	EIEC
	virB	virulence factor gene synthesis,	EIEC
	virF	virulence factor gene expression,	EIEC
	virG	Actin nucleation,	EIEC
	ospB	dampen the inflammatory responses,	EIEC
	ospE	cell detachment,	EIEC
	ospF	dampen the inflammatory responses,	EIEC
	ospG	dampen the inflammatory responses,	EIEC
	ospI	dampen the inflammatory responses,	EIEC
	ospZ	dampen the inflammatory responses,	EIEC
	icsB	autophagy inhibition,	ETEC
	aaiA-Y	type VI secretion system (T6SS),	EAEC
	ibeA	invasion protein ibeA,	AIEC

## Data Availability

We confirm that all data supporting the findings of this study are available within the article.

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
