# Peer review of "Virulence Factors of Enteric Pathogenic *Escherichia coli*: A Review"

_ijms, 2021, doi:10.3390/ijms22189922_

Round 1
Reviewer 1 Report
I have found review interesting however I have two recommendations:
- Please elaborate more precisely on role of enteropathogenic E.coli in human diseases and its consequences. I have found recent article very helpful: Mare, A.D.; Ciurea, C.N.; Man, A.; Tudor, B.; Moldovan, V.; Decean, L.; Toma, F.
Enteropathogenic Escherichia coli—A Summary of the Literature.
Gastroenterol. Insights 2021, 12, 28–40.https://doi.org/10.3390/gastroent
12010004 - What is the most significant change/finding in virulence factors of E.coli since 2016? There are similarities with other articles and I need to clarify specifically importance of this review in 2021.
Author Response
Dear Reviewer 1
- The role of EPEC in human disease and its consequences is more elaborated precisely and revised in the text in the EPEC section (first and last paragraph of the section) using the reference https://doi.org/10.3390/gastroent12010004.
- In this review paper, we have summarized the virulence factor encoding genes until 2016 (which published in some review papers) and have reviewed new trends and recent advances in virulence factors and associated genes of enteric pathogenic E. coli pathotypes published after 2017. We have focused on biofilm formation (e.g., in EPEC and EHEC), new effectors (e.g., in EPEC), toxins (e.g., in ETEC, AIEC) and virulence factors (e.g., in DAEC, AIEC, EIEC) and pathogenesis for all E. coli pathotypes which have been published after 2017. At the present study, we have described the importance of this review (mentioned above) in the last section of the Introduction section. Also, some recent advances are added into EPEC and EHEC sections in this revision. We updated all possible virulence factors and associated genes of enteric pathogenic E. coli pathotypes in this review paper; therefore, this manuscript may be used and considered as a valuable source of virulence factor genes by researchers worldwide who want to investigate the virulence factors of enteric pathogenic E. coli pathotypes by molecular methods.
All references are adjusted and revised accordingly.
Kind regards,
Dr. Wolfram Manuel Brück
Reviewer 2 Report
Summary
This review summarized the main virulence factors found in the 8 enteric pathogenic Escherichia coli pathotypes. The authors provide a broad list of known virulence factors, however much of the pathogenic mechanisms and virulence factors provided have already been detailed in previous review papers (See Croxen et al. 2013. Clin Microbiol Rev. 26:822-80; Croxen & Finlay 2010. Nat Rev Microbiol. 8:26-38; Kaper et al. 2004. Nat Rev Microbiol. 2:123-140). The authors should emphasize what is new and not found in these previous reviews.
Comments
- The main issue with this review is that most of the information can be found in previous reviews on pathogenic E. coli. What new virulence factors have been discovered and what do they mean for understanding pathogenesis of E. coli infections? The manuscript should be revised to address what has not already been reviewed to validate another review on pathogenic E. coli.
- Line 15: “Nine definite enteric E. coli pathotypes have been well characterized”. Then why were only 8 pathotypes discussed?
- Line 38: Although the authors are correct that EHEC can cause HUS affecting the kidneys, EHEC are not classically defined as an ExPEC. There should be clarity that the site of infection is important when differentiating between diarrheagenic and extraintestinal disease.
- Table: Why is there a comma after each activity/function?
- Table: Some virulence factors have been found in more of the pathotypes than listed. For example, toxB has been found in multiple strains of EHEC and EPEC (See Tozzoli et al. 2005 J Clin Microbiol. 43:4052-4056) Each virulence factor should be thoroughly review to include all relevant pathotypes
- Line 154-155: The nomenclature used in this review should be updated to reflect the most up to date information. For example, change “Shiga-like toxin (SLT)” to Shiga toxin (Stx) (See Scheutz et al. 2012. J Clin Microbiol. 50:2951-2963). Capitalize Shiga on Line 155.
- Line 158: Explain why Stx2a, 2c, and 2d are more highly associated with HC and HUS.
- Line 179-180: Is there any evidence that Stx is secreted? Is more research needed in this area?
- Line 185-186: This definition of EHEC should occur at the beginning of the section.
- Line 220: Is there a reference for this citation.
- Should the Table be numbered?
Author Response
Dear Reviewer 2
- In this review paper, we have summarized the virulence factor encoding genes until 2016 (which published in some review papers) and have reviewed new trends and recent advances in virulence factors and associated genes of enteric pathogenic E. coli pathotypes published after 2017. We have focused on biofilm formation (e.g., in EPEC and EHEC), new effectors (e.g., in EPEC), toxins (e.g., in ETEC, AIEC) and virulence factors (e.g., in DAEC, AIEC, EIEC) and pathogenesis for all E. coli pathotypes which have been published after 2017. At the present study, we have described the importance of this review (mentioned above) in the last section of the Introduction section. Also, some recent advances are added into EPEC and EHEC sections in this revision. We updated all possible virulence factors and associated genes of enteric pathogenic E. coli pathotypes in this review paper; therefore, this manuscript may be used and considered as a valuable source of virulence factor genes by researchers worldwide who want to investigate the virulence factors of enteric pathogenic E. coli pathotypes by molecular methods.
- “Nine definite E. coli pathotypes” is changed and revised to “Eight definite enteric E. coli pathotypes” in Line 15 and throughout the manuscript.
- As EHEC are not classically defined as an ExPEC, this statement is removed from the text and revised throughout the manuscript.
- Table: The commas after each activity are removed from the Table and revised and just a comma is added between two or more activities to separate them from each other. Activity/function is changed to Activity in the Table and revised.
- All virulence factors are double checked for each pathotype and revised throughout the Table to include all relevant pathotypes in each virulence factor.
- The nomenclature used in this review are updated. All “Shiga-like toxin” are changed and revised to “Shiga toxin (stx)” throughout the manuscript. All “shiga toxin” are capitalized and revised to “Shiga toxin” throughout the manuscript.
- As recently shown, Stx2a, stx2c, and stx2d positive EHEC isolates are strongly associated with HUS and HC compared to other stx-subgroups and subtypes which are occasionally linked to HUS and HC. This virulence difference is due to the different residues in the binding subunit of the toxins. This explanation is added and revised in the EHEC section of the manuscript.
- This part is revised in the text. “Shiga toxins are released from the bacterial cells through phage-mediated lysis while specific triggers, including SOS inducing agents such as antibiotic therapy or DNA damages, depress the transcription of lytic phase genes. This contributes to the lysis of the STEC bacterial cell and the release of Shiga toxins into the extracellular milieu.” The secretion is removed from the text and this explanation is added into the text to describe the release process of Shiga toxin.
- The first sentence in the lines 185-186 is transferred to the beginning of the section and revised in the text.
- The relevant reference is cited for this part and revised in the text.
- The Table is numbered and revised to Table 1 throughout the manuscript.
All references are adjusted and revised accordingly.
Kind regards,
Dr. Wolfram Manuel Brück
Round 2
Reviewer 2 Report
Summary
This revision has addressed many of the previous comments. There were a few remaining issues to address. Carefully review the entire manuscript to ensure accuracy throughout.
Comments
- Line 150: Define STEC upon first use.
- Line 166-167: To my knowledge, this strain of O104:H4 was confided to Europe or individuals that had travelled to Europe. Indicate the reference for the statement that O104:H4 was spread globally.
- Line 173: Change “SLT” to Shiga toxin.
Author Response
Dear Reviewer 2,
1. Line 150: STEC is defined upon the first use as Shiga toxin-produsing E. coli.
2. Line 166-167: E. coli O104: H4 spread throughout the Europe. This statement is revised in the text and the reference is added.
3. Line 173: "SLT" is revised to Shiga toxin.
Kind regards,
Dr. Wolfram Bruck
